# Adenomyosis in women undergoing hysterectomy for abnormal uterine bleeding associated with uterine leiomyomas

**Neal M. Lonky**[1]*, **Vicki Chiu**[2], **Cecilia Portugal**[2], **Erika L. Estrada**[2], **John Chang**[2], **Heidi Fischer**[2], **Jamie B. Vora**[3], **Lawrence I. Harrison**[1], **Lauren Peng**[4], **Malcolm G. Munro**[5]

**1** Kaiser Permanente Southern California, Orange County, Department of Obstetrics & Gynecology, Anaheim, California, United States of America, **2** Kaiser Permanente Southern California, Department of Research and Evaluation, Pasadena, California, United States of America, **3** AbbVie Inc, North Chicago, Illinois, United States of America, **4** Department of Radiology, Kaiser Permanente, Los Angeles Medical Center, Los Angeles, California, United States of America, **5** Department of Obstetrics and Gynecology, David Geffen School of Medicine at UCLA, Los Angeles, California, United States of America

* neal.m.lonky@kp.org

**Data Availability Statement:** Individual-level data reported in this study involving human research participants are not publicly shared due to potentially identifying or sensitive patient

## Abstract

### Background

Uterine leiomyomas and adenomyosis are both common and often associated with abnormal uterine bleeding (AUB), including the symptom of heavy menstrual bleeding (HMB). Understanding the prevalence of adenomyosis in women with uterine leiomyomas could inform clinicians and patients in a way that may improve therapeutic approaches.

### Objective

To explore the prevalence of adenomyosis in a group of women who underwent hysterectomy for AUB-L, to determine the prevalence of submucous leiomyomas, and to examine the utility of preoperative ultrasound to detect the presence of adenomyosis.

### Methods

The Kaiser Permanente Hysterectomy Database (KPHD) was searched for women aged 18–52 undergoing hysterectomy for leiomyoma-associated chronic AUB (AUB-L) in 2018 and 2019. A target sample of 400 comprised those with at least 3 years in the Health System. Radiologists evaluated preoperative pelvic ultrasound images to determine leiomyoma size and level 2 FIGO type (submucous or other), and the linked electronic medical record abstracted for clinical features, including histopathological evidence of adenomyosis.

### Results

Of the 370 subjects that met the study criteria, adenomyosis was identified via histopathology in 170 (45.9%). There was no difference in the adenomyosis prevalence with (47.1%) and without (43.0%) at least one submucous leiomyoma. Subgroup analysis of ultrasound

information. Upon request, and subject to review, the institutions may provide deidentified aggregate-level data that support the findings of this study. Anonymized data (deidentified data including participant data as applicable) that support the findings of this study may be made available from the investigative team in the following conditions: 1) agreement to collaborate with the study team on all publications, (2) provision of external funding for administrative and investigator time necessary for this collaboration, (3) demonstration that the external investigative team is qualified and has documented evidence of training for human subjects protections, and (4) agreement to abide by the terms outlined in data use agreements between institutions. Interested researchers should contact Kaiser Permanente Southern California of Research & Evaluation Compliance Assurance Services at: ResEvalCAS@kp.org.

**Funding:** Dr. Neal Lonky, Principal Investigator received funding from AbbVie Inc. The funders provided feedback on the scope of work protocol. Contract #00115992.0 The funders had no role in study design, data collection and analysis, decision to publish, or preparation of the manuscript.

**Competing interests:** Dr. Lonky reports a consultancy agreement with AbbVie following conclusion of this research, research support from AbbVie and Merk & Co. is the founder/shareholder of Histologics LLC. Dr Munro reports consultancy agreements with AbbVie Inc, American Regent, Daiichi Sankyo, Hologic Inc, Myovant Inc, Pharmacosmos, and Vifor Pharmaceuticals. He has also received indirect grant support from AbbVie Inc and Pharmacosmos. This does not alter our adherence to PLOS ONE policies on sharing data and materials.

images by an expert radiologist for the presence of adenomyosis demonstrated a positive predictive value of 54.0% and a negative predictive value of 43.4%.

## Conclusions

Adenomyosis was present in almost half of this AUB-L cohort undergoing hysterectomy and was equally prevalent in those with and without submucous leiomyomas as determined by sonographic evaluation. The imaging findings are in accord with prior investigators and demonstrate that 2-D ultrasound is insensitive to the presence of adenomyosis when the uterus is affected by leiomyomas. Further research is necessary to determine the impact of various adenomyosis phenotypes on the presence and severity of the symptom of HMB.

## Introduction

Abnormal uterine bleeding (AUB) is a frequently reported symptom for non-pregnant women in their reproductive years. The prevalence, based on health care system databases, suggests that up to one-third will be affected at some time in their lifetime [1, 2]. While high, these data likely underestimate the true prevalence of these symptoms. Evidence from survey data suggests one of the AUB symptoms, heavy menstrual bleeding (HMB), may have a point prevalence of as high as 50% [3, 4].

Several potential causes or contributors to AUB symptoms have been codified in the system developed by FIGO, known as FIGO AUB System 2, or the PALM-COEIN system, initially published in 2011 [5] and then revised in 2018 [6]. Adenomyosis and leiomyomas are two common findings in women with AUB in the reproductive years. Leiomyomas have been reported in as many as 70 to 80% of women by the age of 50 using simple ultrasound techniques [7]. While at least 50% of affected women are asymptomatic, the overall incidence of AUB associated with leiomyomas (AUB-L) is estimated to be anywhere from 14–25% [5, 8] and it has been estimated that most of the roughly 600,000 hysterectomies performed annually in the US are for one or a combination of AUB and leiomyomas [9, 10]. It is generally accepted that to cause the symptom of heavy menstrual bleeding (HMB), a leiomyoma should be in contact with the endometrium (submucous or SM) [11], thereby presenting an opportunity for the tumor's molecular expressions, such as TGF-$\beta$3, to disrupt local hemostasis [12].

Adenomyosis is defined as the existence of endometrial glands and stroma in the myometrium, typically accompanied by surrounding myometrial hyperplasia and hypertrophy. While adenomyosis is frequently asymptomatic [13] it is another potential cause or contributor to AUB symptoms (AUB-A) with an estimated prevalence, based on imaging studies, of 20–35% [14, 15]. Available evidence suggests that adenomyosis and leiomyomas are commonly found together in women who undergo hysterectomy, typically for AUB, with a reported prevalence ranging from 15 to 57% [16–20].

There is evidence that those women with adenomyosis, in addition to leiomyomas (AUB-A; -L), may have a disproportionate symptom burden, including subjectively increased menstrual bleeding volume and a greater degree of dysmenorrhea [21, 22]. Since both adenomyosis and leiomyomas are frequently asymptomatic, they may coexist with other disorders that are the actual causes or contributors to the AUB symptoms, including coagulopathies (AUB-C), ovulatory dysfunction (AUB-O), and primary endometrial disorders (AUB-E). The advent of various uterine-preserving procedural interventions and the use or introduction of a spectrum of medical interventions challenges clinicians as we enter an era where personalized medicine and shared decision-making are beginning to dominate practice. Consequently,

there exists a need to distinguish amongst these various causes or contributors to AUB and other symptoms in a fashion that informs treatment decisions. While pelvic ultrasound has been shown to be highly sensitive and specific for the detection of adenomyosis [23–25], available evidence suggests that when leiomyomas are present, both sensitivity and specificity diminish [26, 27].

We designed this retrospective, descriptive study to explore the prevalence of adenomyosis in a group of women who underwent hysterectomy for AUB-L, to determine the prevalence of submucous leiomyomas, and to examine the utility of preoperative ultrasound to detect the presence of adenomyosis. The association of histopathological evidence of adenomyosis in cases without a sonographically defined submucous leiomyoma was also examined.

## Methods

### Study design

A retrospective, descriptive, cross-sectional study was designed to query the Hysterectomy Database (HD) populated by a physician entry into the Hysterectomy Registry and associated electronic medical record (EMR) at Kaiser Permanent Southern California (KPSC). In this retrospective data-only study, the research involved minimal risk to participants and involved no procedures for which signed consent is usually required; thus, we received a waiver of written informed consent from the KPSC Institutional Review Board. This study was approved by the Health Maintenance Organization's (HMO) Institutional Review Board, was conducted in accordance with ethical principles of the current Declaration of Helsinki, and was consistent with the International Conference Harmonization Good Clinical Practice (ICH-GCP) and Good Epidemiology Practices (GEP) and applicable regulatory requirements.

### Study population

The source population comprises members of a large integrated healthcare delivery system that provides comprehensive care for over 4.5 million diverse health plan members across southern California. The primary data source for the overall cohort assembly and characterization was from the HMO's Hysterectomy Registry/Database EMR form(s) populating the HD, including the EMR chart data. Part of a systemic policy regarding EMR documentation of benign hysterectomy cases, relevant fields captured in each of the registry entries include pre-operative diagnosis (endometriosis, adenomyosis, and uterine fibroids), history and duration of AUB symptoms, and history and duration of pelvic pain. As part of the operative note, findings are easily captured immediately postoperative with checkboxes documenting the surgeon-observed presence of endometriosis and leiomyoma, among other variables.

The first step was to identify patients aged 18–52 years, with at least 3 years of registration in the HMO, who underwent hysterectomy between January 1, 2018, and December 31, 2019, with a pre-operative (clinical) diagnosis of chronic AUB associated with uterine leiomyomas (AUB-L). To exclude individuals with acute heavy menstrual bleeding, to be eligible, it was necessary to have at least one AUB diagnosis code in the EMR system 30 days before the hysterectomy. The electronic medical record (EMR) was also searched to determine the proportion of cases with an AUB diagnostic code entered 180 or more days before the hysterectomy. This study data collection and analysis was conducted from January 2020- December 2022.

Cases were selected from the registry database if they had a preoperative diagnosis of AUB and leiomyoma and confirmed via chart review documenting the presence of UF along with additional data on fibroid type, location, and size. The cohort of reviewed cases required a post-operative diagnosis of uterine leiomyomas as documented in the related pathology report.

Those with a history of gynecologic malignancy, adnexal mass, or symptomatic pelvic relaxation were excluded from the study.

## Sample determination

A research analyst obtained a stratified random sample of 400 hysterectomy cases from the HD where "leiomyoma" was identified in the preoperative diagnosis field. While random sampling of all age strata was performed and data stratified by age category, it was felt essential to skew the cohort composition such that older women in the later reproductive years were underrepresented. This meant that age strata 40–44 years were randomly selected in a 2:1 radio compared to those aged 45–52 years. All study staff had access to the medical records of the identified cohort from the database for chart abstraction and review of radiology reports. However, during analysis, the data set was aggregated and fully anonymized.

## Chart abstraction

Research Associates conducted chart abstraction tasks utilizing REDCap, a secure web application for building and managing online surveys and databases. The purpose of the chart abstraction was to further assess HD data quality by comparing those data housed in the registry with those from the EMR. The chart abstraction process documented the clinical course of patients with a history of AUB-L who underwent a hysterectomy, including diagnosis, initial symptoms, comorbidities, fertility status, and medical and surgical treatments. The chart abstractors collected relevant data for up to 3 years before the hysterectomy procedure date. The EMR chart review included outpatient, inpatient, and emergency data, pharmacy records, and searchable physician notes. The abstractors also documented the preoperative and postoperative diagnoses and radiologic diagnoses, including ultrasound (US), computerized tomography (CT), and magnetic resonance imaging (MRI). Operative notes and pathology reports associated with the hysterectomy were reviewed to confirm the post-operative diagnosis of uterine leiomyomas and identify endometriosis and/or adenomyosis. The imaging reports were reviewed to determine the size and location (anterior, posterior, lateral, fundal, pedunculated) of the largest fibroids up to three in number.

## Radiologist review

Two health plan radiologists unaware of the final post-hysterectomy pathology diagnosis reviewed images (not reports) after the chart abstractors completed their work to identify missing data on the imaging-based location of the three largest fibroids. Radiologist 1 reviewed images from 231 cases that had missing US report data for the location and diameters of fibroids and recorded these metrics in case report forms. Radiologist 2 was given a unique set of records, also with missing data, and reviewed the US images reporting location, FIGO type, and uterine and leiomyoma volume, and specifically determined if and how many submucous fibroids were found regardless of size. The pelvic ultrasound images of this subset were also reviewed for the presence of adenomyosis using the following criteria: Asymmetrical myometrial thickening, indistinct endo-myometrial interface, echogenic linear striations, nodules extending from the endometrium into the myometrium. The reviewer was asked to select from a scale that ranged from absent features correlating to adenomyosis to unlikely to likely with a category for "can't determine" if the file was unreadable for technical reasons or if the leiomyoma presence obscured interpretation.

## Statistical analysis

The characteristics of the samples of the two radiologists were compared using the χ2 test or the Fisher exact test for categorical variables and the Kruskal–Wallis test for continuous variables, as appropriate. The association between adenomyosis present and absent and FIGO type was assessed using the χ2 test for independence. Two-sided *P* values < 0.05 were considered statistically significant. The sensitivity (percent with radiologically likely adenomyosis who also had histopathologically-determined adenomyosis) and specificity (percent with unlikely or no adenomyosis on imaging who also had histopathological adenomyosis) were calculated to compare the radiologic diagnosis and the gold standard of histopathological examination of the uterus. To further understand these differences, the positive predictive value (PPV) was calculated (percentage with histopathological adenomyosis among those with likely adenomyosis on radiological imaging) as well as the negative predictive value (NPV), the percentage with no histopathological evidence of adenomyosis among those with radiologically unlikely/no adenomyosis. All statistical analyses were performed using SAS version 9.4 (SAS Institute, Inc., Cary, NC).

# Results

## Cohort identification

The hysterectomy database (HD) was searched to identify cases of AUB-L performed on women 18–52 years of age who were in the HMO for at least 3 years before the performance of the procedure. The resulting 1,243 cases were reduced to 400, disproportionately sampling those aged 40–44 years over those 45–52. Of this sample, 14 had no documentation of leiomyomas histopathologically, and 16 had no preoperative ultrasound report in the database (Fig 1).

## Demographics

The subject selection process is summarized. Ultimately, 400 records were obtained for review; after evaluation, 30 were excluded for the absence of fibroids on the pathology report [14] or the pelvic ultrasound report (n = 16), leaving 370 available for analysis. The demographic features of this cohort and the subgroup evaluated by Radiologist 2 are displayed in Table 1. Almost half of the subjects were between 40 and 44, and the mean age was 41.9 years. Notably, nearly 50% of the participants identified as Hispanic, almost a quarter were Black, and just under 20% were White. The mean BMI was 31.4, with 51.8% of these subjects categorized as obese Class 1, 2 or 3. While the inclusion criteria defined the population to include those with an AUB diagnostic code at least 30 days before hysterectomy, a thorough chart audit showed that 277/370 (74.9%) of the cohort had AUB diagnostic codes entered at least 180 days before the surgical intervention.

Table 2 demonstrates the pre-hysterectomy medical and surgical interventions reported for the overall cohort and those with and without adenomyosis. The most common medical interventions were gonadotropin-releasing hormone analogs (36.8%), NSAIDS (62.4%), and progestin therapy, implantable (1.4%), oral (26.2%), or intrauterine (31.6%). Given that D&C and sterilization are not considered therapeutic for AUB, procedural interventions included laparoscopy (15.9%), hysteroscopic myomectomy (9.5%), and myomectomy performed via an abdominal approach (6.8%). There were no differences in the frequency of any pre-hysterectomy intervention between those women with and without histopathological evidence of adenomyosis.

## Evaluation for the presence of adenomyosis and endometriosis

Adenomyosis was identified in 170 of the 370 evaluable cases for a prevalence of 45.9% (Table 1). There was no difference in the histopathological prevalence of adenomyosis based

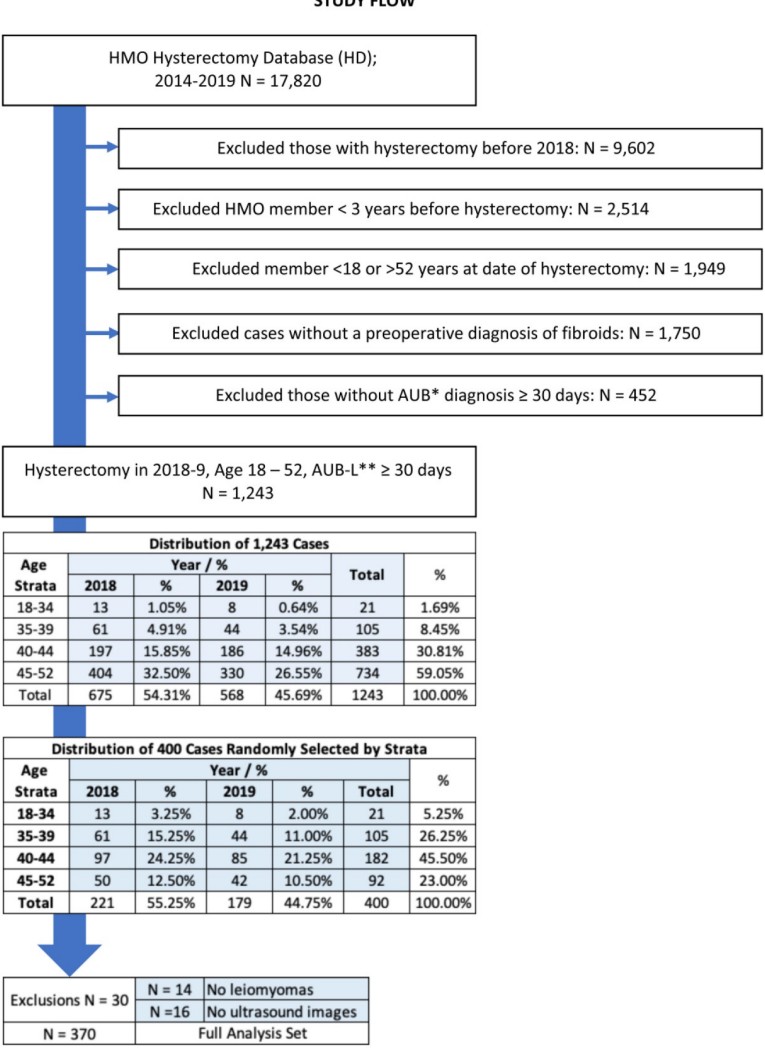

**STUDY FLOW**

**Distribution of 1,243 Cases**

| Age Strata | 2018 | % | 2019 | % | Total | % |
|---|---|---|---|---|---|---|
| 18-34 | 13 | 1.05% | 8 | 0.64% | 21 | 1.69% |
| 35-39 | 61 | 4.91% | 44 | 3.54% | 105 | 8.45% |
| 40-44 | 197 | 15.85% | 186 | 14.96% | 383 | 30.81% |
| 45-52 | 404 | 32.50% | 330 | 26.55% | 734 | 59.05% |
| Total | 675 | 54.31% | 568 | 45.69% | 1243 | 100.00% |

**Distribution of 400 Cases Randomly Selected by Strata**

| Age Strata | 2018 | % | 2019 | % | Total | % |
|---|---|---|---|---|---|---|
| 18-34 | 13 | 3.25% | 8 | 2.00% | 21 | 5.25% |
| 35-39 | 61 | 15.25% | 44 | 11.00% | 105 | 26.25% |
| 40-44 | 97 | 24.25% | 85 | 21.25% | 182 | 45.50% |
| 45-52 | 50 | 12.50% | 42 | 10.50% | 92 | 23.00% |
| Total | 221 | 55.25% | 179 | 44.75% | 400 | 100.00% |

**Fig 1. Adenomyosis in reproductive aged women undergoing hysterectomy for abnormal uterine bleeding associated with uterine leiomyomas.** *AUB: Abnormal uterine bleeding. **AUB-L: Abnormal uterine bleeding associated with leiomyomas.

on the presence or absence of a submucous myoma (Table 3). The presence of endometriosis was documented based either on histopathology or the description in the operative report. Endometriosis was present in 13.5% of the cohort, with no significant differences between those with and without a submucous leiomyoma.

A review of the imaging used by the HMO's clinicians demonstrated that MRI was uncommonly employed, whereas pelvic ultrasound was performed in almost all of the subjects identified in the database. Consequently, the radiologists evaluated pelvic ultrasound images to determine the relevant features of the leiomyomas in this cohort of women. Radiologist 2 reviewed the pelvic ultrasound images from a subgroup of 155 records of subjects with a similar demographic distribution to the overall cohort and a similar histopathological prevalence of adenomyosis (Table 1). Of these, 16 had no identifiable ultrasound images, making 139 evaluable for subgroup analysis (Table 5). A total of 36, or 25.9% of the ultrasound studies,

**Table 1. Demographics of entire cohort and radiological subgroup.**

| | | Adenomyosis Presence | | | | | |
| | | Full Data Set | | | Radiology Subset | | |
| | | Yes | No | Total | Yes | No | Total |
|---|---|---|---|---|---|---|---|
| **Demographic Elements** | | 170 | 200 | 370 | 70 | 69 | 139 |
| Age (year) | Mean±SD | 42.5 ±4.4 | 41.3±4.7 | 41.9±4.6 | 42.7±4.1 | 40.8±4.2 | 41.8±4.3 |
| Age Categories, n (%) | | | | | | | |
| | 18–34 | 4 (2.4) | 15 (7.5) | 19 (5.1) | 0 (0.0) | 6 (8.7) | 6 (4.3) |
| | 35–39 | 41 (24.1) | 54 (27.0) | 95 (25.7) | 18 (25.7) | 19 (27.5) | 37 (26.6) |
| | 40–44 | 80 (47.1) | 89 (44.5) | 169 (45.7) | 35 (50.0) | 35 (50.7) | 70 (50.4) |
| | 45–52 | 45 (26.5) | 42 (21.0) | 87 (23.5) | 17 (24.3) | 9 (13.0) | 26 (18.7) |
| Race Ethnicity, n (%) | | | | | | | |
| | White | 26 (15.3) | 44 (22.0) | 70 (18.9) | 12 (17.1) | 21 (30.4) | 33 (23.7) |
| | Black | 41 (24.1) | 45 (22.5) | 86 (23.2) | 14 (20.0) | 15 (21.7) | 29 (20.9) |
| | Hispanic | 85 (50.0) | 94 (47.0) | 179 (48.4) | 38 (54.3) | 29 (42.0) | 67 (48.2) |
| | Asian/Pacific Islander | 15 (8.8) | 15 (7.5) | 30 (8.1) | 5 (7.1) | 4 (5.8) | 9 (6.5) |
| | Other/Unknown | 3 (1.8) | 2 (1.0) | 5 (1.4) | 1 (1.4) | 0 (0.0) | 1 (0.7) |
| BMI (kg/m$^2$) | Mean±SD | 31.0±6.5 | 31.7±6.7 | 31.4±6.6 | 31.3±7.2 | 31.2±6.6 | 31.2±6.9 |
| BMI Categories, n (%) | | | | | | | |
| | < 24.9 (under/normal) | 29 (17.1) | 36 (18.0) | 65 (17.6) | 13 (19.6) | 16 (23.2) | 29 (20.9) |
| | 25.0–29.9 (overweight) | 52 (30.6) | 59 (29.5) | 111 (30.0) | 21 (30.0) | 17 (24.6) | 38 (27.3) |
| | 30.0–34.9 (obese class 1) | 46 (27.1) | 47 (23.5) | 93 (25.1) | 18 (25.7) | 16 (23.2) | 34 (24.5) |
| | 35.0–39.9 (obese class 2) | 27 (15.9) | 33 (16.5) | 60 (16.2) | 10 (14.3) | 13 (18.8) | 23 (16.6) |
| | 40+ (obese class 3) | 15 (8.8) | 24 (12.0) | 39 (10.5) | 7 (10.0) | 7 (10.1) | 14 (10.1) |
| | Missing | 1 (0.6) | 1 (0.5) | 2 (0.5) | 1 (1.4) | 0 (0.0) | 1 (0.7) |

were designated "can't determine" by the radiologist. This left 103 subjects for which the reviewer determined the presence or absence of sonographic features suggestive of adenomyosis (Table 4). The categories "no" and "unlikely" were conflated to indicate that such features were not present, while 'likely' suggested the presence of adenomyosis. Using these categories, the sensitivity and specificity were 47.4% and 50.0%, respectively, while the positive and negative predictive values were 54.0 and 43.4%.

Table 5 demonstrates the relationship between adenomyosis and a preoperative complaint of either AUB only or AUB and chronic pelvic pain. The chronic pelvic pain elements comprised one or a combination of dysmenorrhea, dyspareunia, and acyclic pelvic pain or pressure. Those women with pain and AUB were no more likely to have adenomyosis identified in their hysterectomy specimens.

## Discussion

The data obtained from the present study showing 45.9% with histological evidence of adenomyosis are consistent with the notion that the prevalence of adenomyosis in those undergoing hysterectomy for benign reasons is high. Our 2-D ultrasound data demonstrated that 71% of the cohort had findings consistent with at least one submucous leiomyoma, including the spectrum of FIGO Types, including types 2–5, 3–5, and 3. There was no difference in the frequency of histologically-determined adenomyosis based on the presence or absence of at least one submucous leiomyoma.

The study expert radiologist could only offer an opinion regarding the presence or absence of adenomyosis in three-quarters of the 2-D ultrasound images. Even in those cases where an

**Table 2. Pre-hysterectomy medical and procedural interventions.**

| | Adenomyosis | | | |
| --- | --- | --- | --- | --- |
| | **Yes (N = 170)** | **No (N = 200)** | **Total (N = 370)** | **P-value[1]** |
| **Pre-surgical treatment, n (%)** | | | | |
| GnRH analog | 59 (34.7) | 77 (38.5) | 136 (36.8) | 0.5164 |
| Combined Hormonal Contraception | | | | |
| Oral | 58 (34.1) | 80 (40.0) | 138 (37.3) | 0.2809 |
| NuvaRing | 1 (0.6) | 3 (1.5) | 4 (1.1) | 0.6279 |
| Progestins & Progestin Modulators | | | | |
| Nexplanon/Implanon | 2 (1.2) | 3 (1.5) | 5 (1.4) | 1.0000 |
| Progesterone Receptor Blocker | 1 (0.6) | 3 (1.5) | 4 (1.1) | 0.6279 |
| Progestin Containing IUS (Mirena) | 48 (28.2) | 69 (34.5) | 117 (31.6) | 0.2179 |
| Oral or injectable | 41 (24.1) | 56 (28.0) | 97 (26.2) | 0.4092 |
| Tranexamic Acid | 7 (4.1) | 9 (4.5) | 16 (4.3) | 1.0000 |
| NSAIDS | 114 (67.1) | 117 (58.5) | 231 (62.4) | 0.1062 |
| Other | 48 (28.2) | 44 (22.0) | 92 (24.9) | 0.1851 |
| None | 10 (5.9) | 14 (7.0) | 24 (6.5) | 0.8328 |
| **Surgical procedure before hysterectomy, n (%)** | | | | |
| **Myomectomy** | | | | |
| Hysteroscopic | 15 (8.8) | 20 (10.0) | 35 (9.5) | 0.7253 |
| Laparoscopic or laparotomic | 9 (5.3) | 16 (8.0) | 25 (6.8) | 0.4064 |
| Uterine artery embolization | 2 (1.2) | 1 (0.5) | 3 (0.8) | 0.5960 |
| Laparoscopy (Diagnostic Only) | 1 (0.6) | 4 (2.0) | 5 (1.4) | 0.3799 |
| Laparoscopy (Operative) | 21 (12.4) | 38 (19.0) | 59 (15.9) | 0.0887 |
| Endometrial ablation | 10 (5.9) | 12 (6.0) | 22 (5.9) | 1.0000 |
| High-Frequency Ultrasound (HiFU) | 1 (0.6) | 0 (0.0) | 1 (0.3) | 0.4595 |
| Hysteroscopic resection (Not specified) | 16 (9.4) | 14 (7.0) | 30 (8.1) | 0.4473 |
| Leiomyoma RF Ablation | 1 (0.6) | 0 (0.0) | 1 (0.3) | 0.4595 |
| Ovarian cystectomy | 6 (3.5) | 4 (2.0) | 10 (2.7) | 0.5227 |
| Removal of ovary | 5 (2.9) | 2 (1.0) | 7 (1.9) | 0.2548 |
| Salpingectomy | 11 (6.5) | 5 (2.5) | 16 (4.3) | 0.0743 |
| Tubal ligation/sterilization/Essure | 43 (25.3) | 34 (17.0) | 77 (20.8) | 0.0547 |

[1]Fisher Exact p-value;

opinion was offered, the sensitivity and specificity were in the range of 50%, data that suggest that, at least in this study, transvaginal ultrasound was not a useful tool for detecting adenomyosis women with leiomyomas.

Other investigators have found that adenomyosis is frequently found in women undergoing hysterectomy for benign indications, with prevalence ranging from 8.8 to 61.5% [16–19, 28]. Sonographic evidence of adenomyosis in the presence of leiomyomas has been previously identified in 22.8% of women attending a UK gynecological clinic [14]. However, to our knowledge, this is the first report specifically evaluating the prevalence of adenomyosis in a population of women undergoing hysterectomy for chronic AUB-L. While the prevalence of 45.9% is high, the lack of a standardized pathological protocol for uterine dissection leaves open the possibility that the actual prevalence of adenomyosis in this population may be even higher, as suggested by Bird et al. [28].

Using our methodology, the prevalence of endometriosis in this cohort was 13.5%: 15.3% in cases where adenomyosis was identified. Evidence on the combined prevalence of

Table 3. Presence of adenomyosis and endometriosis by leiomyoma subtype[1].

| Leiomyoma Type (FIGO Level 2) | Subtype (Radiology) N = 370 | Comorbidity (Pathology and/or OR Report) | | | |
|---|---|---|---|---|---|
| | | Adenomyosis +/- Endometriosis | Leiomyoma +/- Endometriosis | Total | Adenomyosis |
| | | N | N | N | % |
| Submucous n = 263 | Intracavitary (0) | 4 | 2 | 6 | 66.7% |
| | Submucous (1, 2) | 21 | 17 | 38 | 55.3% |
| | Intramural/Submucous (3) | 34 | 31 | 65 | 52.3% |
| | Trans-mural (2–5, 3–5) | 65 | 89 | 154 | 42.2% |
| | **Subtotal** | **124** | **139** | **263** | **47.1%** |
| Other n = 107 | Intramural (4) | 22 | 18 | 40 | 55.0% |
| | Subserous (5, 6) | 19 | 31 | 50 | 38.0% |
| | Pedunculated (7) | 5 | 12 | 17 | 29.4% |
| | **Subtotal** | **46** | **61** | **107** | **43.0%** |
| **TOTAL** | | **170** | **200** | **370** | **45.9%** |

[1]Based on FIGO Level 2 criteria (submucous or "other").

endometriosis and adenomyosis is not robust [29] with available estimates ranging from a third to more than half depending on the phenotypes of endometriosis and adenomyosis studied [30, 31]. We believe that the prevalence of endometriosis in our cohort may be an underestimation. This relates, in part, to the fact that in this retrospective work we were able to confirm the presence of adenomyosis and leiomyomas histopathologically, while the diagnosis of endometriosis was largely based upon the operative reports that were created outside of any research protocol.

It is unclear to what degree adenomyosis added to this cohort's pain and AUB symptoms that led to the hysterectomy. However, given the presumption that submucous leiomyomas are more likely to cause the symptom of HMB, we hypothesized that women without such lesions would have a higher prevalence of adenomyosis. However, this was not the case, as adenomyosis was found histopathologically in 46.3% of those cases with a submucous leiomyoma and 45.2% without such a finding. While there are a number of possible explanations for such findings, it is apparent that adenomyosis, like many disorders, comprises a spectrum of phenotypical and molecular expressions that manifest in a variable degree of symptoms. For example, using transvaginal ultrasound for the diagnosis of adenomyosis, and pictorial blood loss

Table 4. Radiologist versus histopathological diagnosis of adenomyosis[1].

| Radiologist Read | Adenomyosis (final post-operative diagnosis) | | |
|---|---|---|---|
| | Present | Absent | Total |
| Likely | 27 | 23 | **50** |
| No/Unlikely | 30 | 23 | **53** |
| **Total** | **57** | **46** | 103 |
| Sensitivity | 47.4% | | |
| Specificity | 50.0% | | |
| PPV | 54.0% | | |
| NPV | 43.4% | | |

[1]Only 74.1% of these images were evaluable by the radiologist for the presence of adenomyosis, so the utility of these ultrasound images was even less than reflected in the table data.

**Table 5. Presence of pain[1].**

| FIGO Type | Adenomyosis Present | | Adenomyosis Absent | | Total |
|---|---|---|---|---|---|
| | N | % | N | % | |
| 0 or 1 | 1 | 1.4 | 0 | 0.0 | 1 |
| 1 | 6 | 8.6 | 4 | 5.8 | 10 |
| **Both 0 & 1** | **7** | **10.0** | **4** | **5.8** | **11** |
| 2 | 8 | 11.4 | 5 | 7.2 | 13 |
| 2–5 | 32 | 45.7 | 37 | 53.6 | 69 |
| **Both 2 & 2–5** | **40** | **57.1** | **42** | **60.9** | **82** |
| 3 | 5 | 7.1 | 2 | 2.9 | 7 |
| 3–5 | 1 | 1.4 | 0 | 0.0 | 1 |
| **Both 3 & 3–5** | **6** | **8.6** | **2** | **2.9** | **8** |
| **Any Submucous** | **53** | **75.7** | **48** | **69.6** | **101** |
| **No Submucous** | **17** | **24.3** | **21** | **30.4** | **38** |
| **Total** | **70** | **100.0** | **69** | **100.0** | **139** |

[1]Comparison of the diagnosis of adenomyosis in women presenting with AUB only and AUB with chronic pelvic pain.

assessment chart (PBLAC) scores [32] for menstrual volume, Naftalin and colleagues demonstrated that the mean scores varied from normal (<100), to about 300 depending on the number of sonographically visualized adenomyosis features [33]. Others have reported on the histopathological evaluation of hysterectomy specimens for depth of involvement and the number of foci of glandular tissue in the myometrium and found associations with the estimated volume of bleeding [28, 34]. Absent a prospective protocol, we were unable to characterize the adenomyosis in this cohort beyond determining its presence or absence. There are also several other possible causes of the symptom of HMB that were not addressed in this work. These include non-structural causes of AUB, such as coagulopathies, ovulatory dysfunction, and primary disorders of endometrial hemostasis, diagnoses that were not consistently evaluable in the health plan's Hysterectomy Database or EMR used as source materials for this study.

The use of transvaginal pelvic ultrasound and MRI-based uterine imaging for the diagnosis of adenomyosis has revolutionized the evaluation of a disorder that was previously confirmed only following hysterectomy [27, 35–37]. High-quality evidence has suggested that two-dimensional (2-D) ultrasound is not only highly specific and sensitive for the diagnosis of adenomyosis but is substantially equivalent to MRI as each has sensitivity and specificity in the range of 80% [23–25] that may be even higher with newer techniques, the inclusion of more features, and, for ultrasound, performed and interpreted by a well-trained sonographer [37, 38]. However, it should be understood that the evidence regarding the sensitivity and specificity of transvaginal ultrasound was acquired principally from studies evaluating uteri unaffected with leiomyomas. The relatively poor sensitivity and specificity for 2-D transvaginal ultrasound found in our study supports the work of other investigators [26, 27]. Consequently, MRI appears to be a better imaging method when it is deemed important to evaluate a uterus for adenomyosis when leiomyomas are present.

For clinicians, determining the etiology and, thus, a plan for management for women with chronic AUB can be complicated. In part to address this issue, a nomenclature system to describe AUB systems (FIGO AUB System 1) and an etiology-based classification system were first designed and published by the International Federation of Gynecology and Obstetrics

(FIGO) in 2011 [5] and revised in 2018 [6]. The classification system, called System 2, and known by the acronym "PALM-COEIN", categorizes structural abnormalities of the endometrium and myometrium as well as non-structural local and systemic disorders that impact endometrial hemostasis including coagulopathies (AUB-C), ovulatory disorders (AUB-O), and primary endometrial dysfunction (AUB-E). It is recognized that one or more of these pathologies can co-exist. Still, and importantly, many structural abnormalities are asymptomatic–including polyps, adenomyosis, and leiomyomas–leaving the functional disorders as the actual cause of the patient's complaint. Currently, evaluating individuals for these nonstructural causes of AUB in general and HMB, in particular, requires the implementation of FIGO AUB System 1, a structured history that includes the determination of ovulatory status and risk factors for the presence of a coagulopathy. Unfortunately, the design of our protocol precluded any determination of these historical features and potential diagnoses of AUB-C, -O, and -E.

Although it seems clear that some leiomyomas cause or contribute to AUB symptoms in general and that of HMB in particular, it is also apparent that not all leiomyomas cause abnormal "menstrual" bleeding [39, 40]. By extension, it can be hypothesized that when AUB exists in the presence of leiomyomas (AUB-L), there may be other causes or contributors to the symptoms that include endometrial polyps or adenomyosis and the non-structural disorders that are associated with deficient systemic or local hemostatic mechanisms. Consequently, procedures such as endometrial polypectomy or even myomectomy may not improve the bleeding symptoms, and an observation clinicians should consider and share with patients contemplating surgical management.

This work's findings, strengths, and limitations can inform the design and interpretation of future research. First, the utility of such large databases would be well served by more granular inclusion of specific symptoms and findings related to abnormal uterine bleeding and chronic pain in general and adenomyosis and leiomyomas in particular. The two FIGO AUB Systems [6] could form the core of abnormal uterine bleeding databases, and elements such as those based on work by members of the International Pelvic Pain Society would be useful for chronic pain [41]. The asymptomatic nature of many cases of endometriosis and uterine findings, including leiomyomas and endometriosis, means that it is important for databases to capture other potential contributors to AUB, such as coagulopathies (AUB-C), ovulatory dysfunction (AUB-O), and primary endometrial disorders (AUB-E) as well as the spectrum of potential causes of chronic pelvic pain such as levator floor myalgia, irritable bowel syndrome, and bladder dysfunction. Such an approach is also important for the results of pelvic imaging included in such databases, with standardized reporting of ultrasound and MRI findings. Of course. There is a need for the development of genetic and molecular markers that may assist in determining the clinical relevance of findings demonstrated by the clinical assessment.

The strengths of this retrospective study include the unique perspective of evaluating women undergoing hysterectomy for AUB-L and the relatively large sample size taken from the database of a community health care system. These circumstances may support the generalizability of the results. The preoperative diagnosis entered into the HD was remarkably accurate in predicting the presence of leiomyomas in this population. Still, it was less robust as a tool for identifying adenomyosis and the various nonstructural potential contributors to AUB symptoms. This is due to the fact it was a database linked to the hysterectomy procedure and not to a granular repository of clinical features that would allow a more precise determination of the symptoms, findings, and appropriate laboratory investigations necessary to identify the nonstructural causes of chronic AUB.

There are several other limitations to this work. The prevalence of adenomyosis was high and consistent with the reports of others but may have been underestimated because there was

no prospective protocol for dissection of the myometrium. Due to the retrospective design, slide review was not consistently available, and specimens were discarded, not availing the investigators' availability for extra sectioning, likely down biasing the prevalence of adenomyosis. The database was not designed to provide granularity regarding elements of patient symptomatology, including the frequency, duration, regularity, and subjectively determined severity of menstrual blood loss, features that might give some indication of the presence of other causes or contributors to AUB symptoms. Other investigators have associated more severe pelvic pain in women with leiomyomas and adenomyosis [21, 22]. However, the database was also limited with respect to characteristics of the pain experienced by the subjects, including cyclicity, severity, duration, and location that could have contributed to the decision to undergo hysterectomy. The radiologists comprised a single generalist and one with specific interest and training in the interpretation of pelvic ultrasounds. A larger cohort of radiologists with different training and in other institutions might have provided more generalizable results.

## Conclusion

This work is consistent with the results of others demonstrating that adenomyosis is frequently found in women deciding to undergo hysterectomy for AUB-L. However, the role of adenomyosis in the generation of AUB symptoms, including HMB, is unclear from these data. Histopathological or imaging-based, well-defined phenotyping for disease burden might identify features more likely to contribute to symptoms. Similar to the findings of other investigators, 2-D pelvic ultrasound appears to be of limited value for the diagnosis of adenomyosis in the presence of leiomyomas: MRI is probably more precise and sensitive but was uncommonly utilized and, consequently, not evaluated in this work. Nevertheless, if there is suspicion that adenomyosis may be contributing to the patient's symptoms, MRI should be considered as a more appropriate evaluation method in a way that better informs counseling regarding medical or surgical interventions. Furthermore, to be optimally useful, databases such as the HD would benefit from the inclusion of clinical features assimilated in a fashion that facilitates a more complete exploration of the contributors to AUB symptoms. Such an approach could facilitate the research necessary for clinicians to design more personalized treatment strategies.

## Author Contributions

**Conceptualization:** Neal M. Lonky, Jamie B. Vora, Malcolm G. Munro.

**Data curation:** Vicki Chiu, Cecilia Portugal, Erika L. Estrada, John Chang, Heidi Fischer, Lawrence I. Harrison, Lauren Peng.

**Formal analysis:** Heidi Fischer.

**Funding acquisition:** Neal M. Lonky, Malcolm G. Munro.

**Methodology:** Neal M. Lonky, Vicki Chiu, Cecilia Portugal, Heidi Fischer, Jamie B. Vora, Lauren Peng, Malcolm G. Munro.

**Supervision:** Neal M. Lonky, Cecilia Portugal, Malcolm G. Munro.

**Validation:** Neal M. Lonky, Malcolm G. Munro.

**Writing – original draft:** Neal M. Lonky, Vicki Chiu, Heidi Fischer, Malcolm G. Munro.

**Writing – review & editing:** Neal M. Lonky, Vicki Chiu, Cecilia Portugal, Erika L. Estrada, John Chang, Heidi Fischer, Malcolm G. Munro.

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
