## [Decision Letter · Decision Letter 0]

18 Aug 2023

PONE-D-23-09324Adenomyosis in Women Undergoing Hysterectomy for Abnormal Uterine Bleeding Associated with Uterine LeiomyomasPLOS ONE

Dear Dr. Lonky,

Thank you for submitting your manuscript to PLOS ONE. After careful consideration, we feel that it has merit but does not fully meet PLOS ONE’s publication criteria as it currently stands. Therefore, we invite you to submit a revised version of the manuscript that addresses the points raised during the review process.

We look forward to receiving your revised manuscript.

Kind regards,

Alessandro Favilli, PhD, MD

Academic Editor

PLOS ONE

Journal Requirements:

"Dr. Neal Lonky, Principal Investigator received funding from AbbVie Inc. The funders provided feedback on the scope of work protocol. Contract #00115992.0"

3. Thank you for stating the following in the Competing Interests/Financial Disclosure * (delete as necessary) section:

"This manuscript is original research and the authors of this manuscript, and financial disclosures include the following:

o             Dr. Lonky reports a consultancy agreement with AbbVie following conclusion of this research, research support from AbbVie and Merk & Co. is and is the founder/shareholder of Histologics LLC.

o             Dr Munro reports consultancy agreements with AbbVie Inc, American Regent, Daiichi Sankyo, Hologic Inc, Myovant Inc, Pharmacosmos, and Vifor Pharmaceuticals. He has also received indirect grant support from AbbVie Inc and Pharmacosmos."

We note that you received funding from a commercial source:"AbbVie and Merk & Co, American Regent, Daiichi Sankyo, Hologic Inc, Myovant Inc, Pharmacosmos, and Vifor Pharmaceuticals"

4. Please amend the manuscript submission data (via Edit Submission) to include author "Malcolm G. Munro"

Reviewers' comments:

Reviewer's Responses to Questions

**Comments to the Author**

1. Is the manuscript technically sound, and do the data support the conclusions?

Reviewer #1: Yes

2. Has the statistical analysis been performed appropriately and rigorously? 

Reviewer #1: Yes

3. Have the authors made all data underlying the findings in their manuscript fully available?

Reviewer #1: Yes

4. Is the manuscript presented in an intelligible fashion and written in standard English?

Reviewer #1: Yes

5. Review Comments to the Author

Reviewer #1: thank you to giving me the chance to reviw this paper.

despite good overall merit, I have some comments:

-please discuss on relevant literature on the prevalence of adenomyosis in women with endometriosis, comparing your data with those from these letters (eg doi 10.1016/j.ejogrb.2014.08.016)

- US diagnosis of adenomyosis is difficult also for expert operators, many efforts should be done to optimize its diagnostic performance and gain standardization of diagnostic criteria; discuss on novel findings on these topics (eg doi 10.3390/jpm12101572; PMID: 36767092

6. PLOS authors have the option to publish the peer review history of their article (what does this mean?). If published, this will include your full peer review and any attached files.

Reviewer #1: No

---

## [Author Response · Author response to Decision Letter 0]

29 Sep 2023

Dear Editor, 

 We thank you and the reviewer for the positive feedback and the constructive comments provided. Our responses to reviewer comments are listed below and the changes made in the manuscript are tracked with tracked change. We appreciate the opportunity to revise and resubmit our manuscript and look forward to hearing back from PLOS ONE.

Editor’s Specific Comments: 

Response: We have addressed any style requirements, including file naming using the PLOS ONE style templates. 

"Dr. Neal Lonky, Principal Investigator received funding from AbbVie Inc. The funders provided feedback on the scope of work protocol. Contract #00115992.0"

Response: The funders had no role in study design, data collection and analysis, decision to publish, or preparation of the manuscript. 

3. Thank you for stating the following in the Competing Interests/Financial Disclosure * (delete as necessary) section:

"This manuscript is original research and the authors of this manuscript, and financial disclosures include the following:

o Dr. Lonky reports a consultancy agreement with AbbVie following conclusion of this research, research support from AbbVie and Merk & Co. is and is the founder/shareholder of Histologics LLC.

o Dr Munro reports consultancy agreements with AbbVie Inc, American Regent, Daiichi Sankyo, Hologic Inc, Myovant Inc, Pharmacosmos, and Vifor Pharmaceuticals. He has also received indirect grant support from AbbVie Inc and Pharmacosmos."

We note that you received funding from a commercial source:"AbbVie and Merk & Co, American Regent, Daiichi Sankyo, Hologic Inc, Myovant Inc, Pharmacosmos, and Vifor Pharmaceuticals"

Response: Please see the amended Competing Interests Statement below: 

Dr. Lonky reports a consultancy agreement with AbbVie following conclusion of this research, research support from AbbVie and Merk & Co. is the founder/shareholder of Histologics LLC.Dr Munro reports consultancy agreements with AbbVie Inc, American Regent, Daiichi Sankyo, Hologic Inc, Myovant Inc, Pharmacosmos, and Vifor Pharmaceuticals. He has also received indirect grant support from AbbVie Inc and Pharmacosmos. This does not alter our adherence to PLOS ONE policies on sharing data and materials.

4. Please amend the manuscript submission data (via Edit Submission) to include author "Malcolm G. Munro"

Response: We amended the manuscript submission data to include author “Malcolm G. Munro.” 

Response: Please see our Data Availability statement below: 

Individual-level data reported in this study involving human research participants are not publicly shared due to potentially identifying or sensitive patient information. Upon request, and subject to review, the institutions may provide deidentified aggregate-level data that support the findings of this study. Anonymized data (deidentified data including participant data as applicable) that support the findings of this study may be made available from the investigative team in the following conditions: 1) agreement to collaborate with the study team on all publications, (2) provision of external funding for administrative and investigator time necessary for this collaboration, (3) demonstration that the external investigative team is qualified and has documented evidence of training for human subjects protections, and (4) agreement to abide by the terms outlined in data use agreements between institutions. Interested researchers should contact Kaiser Permanente Southern California of Research & Evaluation Compliance Assurance Services at: ResEvalCAS@kp.org.

Reviewer #1: 

Thank you to giving me the chance to review this paper.

despite good overall merit, I have some comments:

-please discuss on relevant literature on the prevalence of adenomyosis in women with endometriosis, comparing your data with those from these letters (eg doi 10.1016/j.ejogrb.2014.08.016).

Response: We have added our data on the prevalence of endometriosis in our cohort, as well as appropriate citations. We do emphasize that our study design was such that histopathological diagnosis was minimally contributory while we had to rely upon operative reports to determine whether or not endometriosis was present at the time of surgery. Because there was no structured method for entering these data, the prevalence of endometriosis in this cohort is likely underreported.

- US diagnosis of adenomyosis is difficult also for expert operators, many efforts should be done to optimize its diagnostic performance and gain standardization of diagnostic criteria; discuss on novel findings on these topics (eg doi 10.3390/jpm12101572; PMID: 36767092

Response: We thank the reviewer for this suggestion but feel that discussion of the standardization of diagnostic criteria is beyond the scope of our work. In this study we asked an experienced radiologist to retrospectively evaluate ultrasounds for the presence or absence of adenomyosis but did not evaluate by criteria. We have dealt with our findings in the context of what is known, particularly about the sensitivity of ultrasound for adenomyosis in the presence of uterine leiomyomas. starting in lines 397-419.

---

## [Editor Report · Decision Letter 1]

13 Nov 2023

Adenomyosis in women undergoing hysterectomy for abnormal uterine bleeding associated with uterine leiomyomas

PONE-D-23-09324R1

Dear Dr. Lonky

We’re pleased to inform you that your manuscript has been judged scientifically suitable for publication and will be formally accepted for publication once it meets all outstanding technical requirements.

Kind regards,

Alessandro Favilli, PhD, MD

Academic Editor

PLOS ONE

Additional Editor Comments (optional):

I would like to express my appreciation for your efforts in addressing the reviewers' suggestions and making the necessary improvements.

The manuscript has now ready to be accepted for publication.

The revisions you made have significantly strengthened the overall quality of the paper, and it aligns well with the standards of our journal.

I understand that the review process may have taken longer than usual, and I sincerely apologize for any inconvenience this may have caused.
---

## [Editor Report · Acceptance letter]

1 Dec 2023

PONE-D-23-09324R1 

Adenomyosis in women undergoing hysterectomy for abnormal uterine bleeding associated with uterine leiomyomas 

Dear Dr. Lonky:

I'm pleased to inform you that your manuscript has been deemed suitable for publication in PLOS ONE. Congratulations! Your manuscript is now with our production department. 

Kind regards, 

on behalf of

Dr. Alessandro Favilli 

Academic Editor

PLOS ONE